# Resistant Starch as a Dietary Intervention to Limit the Progression of Diabetic Kidney Disease

**DOI:** 10.3390/nu14214547

**Published:** 2022-10-28

**Authors:** Anna M. Drake, Melinda T. Coughlan, Claus T. Christophersen, Matthew Snelson

**Affiliations:** 1Glycation, Nutrition and Metabolism Laboratory, Department of Diabetes, Central Clinical School, Monash University, Melbourne 3004, Australia; 2Baker Heart & Diabetes Institute, Melbourne 3004, Australia; 3School of Medical and Health Sciences, Edith Cowan University, Joondalup 6027, Australia; 4WA Human Microbiome Collaboration Centre, School of Molecular Life Sciences, Curtin University, Bentley 6102, Australia

**Keywords:** diabetic kidney disease, diabetes, diet, resistant starch, high-amylose maize starch, gut microbiota, short chain fatty acids

## Abstract

Diabetes is the leading cause of kidney disease, and as the number of individuals with diabetes increases there is a concomitant increase in the prevalence of diabetic kidney disease (DKD). Diabetes contributes to the development of DKD through a number of pathways, including inflammation, oxidative stress, and the gut-kidney axis, which may be amenable to dietary therapy. Resistant starch (RS) is a dietary fibre that alters the gut microbial consortium, leading to an increase in the microbial production of short chain fatty acids. Evidence from animal and human studies indicate that short chain fatty acids are able to attenuate inflammatory and oxidative stress pathways, which may mitigate the progression of DKD. In this review, we evaluate and summarise the evidence from both preclinical models of DKD and clinical trials that have utilised RS as a dietary therapy to limit the progression of DKD.

## 1. Introduction

Diabetes Mellitus (DM) represents a significant health issue as its prevalence continues to increase globally [1]. A large proportion of the burden of disease imposed by DM is a result of its long-term complications [2] (Figure 1), with diabetic kidney disease (DKD) being the DM complication associated with the greatest physical and financial costs [3,4]. DKD develops in approximately 20–40% of patients with diabetes [5,6] and is the leading cause of chronic kidney disease (CKD) and end stage kidney disease (ESKD) world-wide [6,7,8]. In 2014, an estimated 250,000 Australians had DKD, a figure that is expected to exceed 500,000 by 2025 [6] as cases of DM continue to rise [5,6]. Additionally, Australian figures show that from 2013 to 2018 the percentage of new ESKD diagnoses attributed to DKD rose from 26% to 38% [9,10]. DKD is an especially significant issue within Australian Indigenous communities [11], who are five times more likely to report DKD as compared to non-Indigenous Australians [12]. Currently there is a considerable lack of clinically effective interventions to prevent the progression of this condition [13]. As cases of DKD are only expected to rise in reflection of growing DM rates [5,6], it is imperative to explore new management strategies to reduce future burden for both patients and healthcare systems [3,6,8]. Given the progressive nature of DKD, prevention and early intervention present the most promising options to limit the effects of this condition [3,4,7,14].

## 2. Pathogenesis of DKD

The pathogenesis of DKD is complex and multifaceted, and remains poorly understood despite continual advancements in disease understanding [15,16]. Currently established mechanisms driving disease development include hyperglycaemia, altered haemodynamics and hyperlipidaemia, however there is emerging evidence to support the role of perturbations in the gut-kidney axis in this pathogenic process. These forces promote and work alongside a number of inflammatory molecules and mechanisms to create the chronic subacute inflammatory state that underpins DKD development [16,17,18] (Figure 2).

### 2.1. Hyperglycaemia

Hyperglycaemia is a key driving force in DKD pathogenesis [19]. Many cells within the kidney are particularly susceptible to the effects of glucose [2]. In particular, mesangial cells and proximal tubular epithelial cells have limited ability to downregulate glucose transport across their cell membrane [20]. Therefore, in the setting of hyperglycaemia, these cells are exposed to an unregulated influx of glucose into their intracellular space [21]. Raised intracellular glucose concentrations result in increased mitochondrial production of superoxide, a reactive oxygen species (ROS) [22]. It is widely believed that this ‘single process underlies different hyperglycaemia-induced pathogenic mechanisms’ [22] as superoxide inhibits the enzyme GAPDH which is the rate limiting step in the glycolysis pathway [23]. Interruption to this pathway leads to an accumulation of glucose, glucose 6-P, fructose 6-P and glyceraldehyde 3-P which are subsequently rediverted and utilised by other metabolic processes [23]. This results in decreased antioxidant production [24] and increased production of Tumour Necrosis Factor alpha (TNF-α), Transforming Growth Factor beta (TGF-β) [25], advanced glycation endproducts (AGEs) and Protein Kinase C (PKC) [26]. PKC is an enzyme that promotes TGF-β [23,27], ROS production [23] and Angiotensin 2 (Ang2) [27]. The role of hyperglycaemia in early stage DKD development is further evidenced as intensive glucose management to keep fasting plasma glucose (FPG) below 6 mmol/L reduces the risk of albuminuria over 10 years by 34% in newly diagnosed T2DM patients [28].

### 2.2. Haemodynamic Factors

In DKD, increased intraglomerular blood pressure promotes glomerular hyperfiltration [5], an early sign of disease that corresponds with a brief compensatory increase in estimated glomerular filtration rate (eGFR) [5]. Continued exposure to these high pressures exerts significant mechanical stress on renal vascular endothelial cells. This leads to a state of endothelial dysfunction that promotes inflammation and ROS production [24,29]. The main mechanisms responsible for this increase in intraglomerular pressure are systemic hypertension and the upregulation of the Renin Angiotensin Aldosterone System (RAAS) [5,30]. The RAAS is a homeostatic feedback loop that in normal conditions acts to regulate systemic blood pressure [2]. A key hormone in this system, Ang2, exerts its effects primarily in the kidneys and is observed to be increased in DKD [30,31]. Ang2 raises glomerular pressure through vasoconstriction of the efferent renal arteriole. Ang2 also promotes renal inflammation and fibrosis through activating TGF-β [32,33] and Nuclear Factor Kappa B (NF-κB) [34,35], and upregulating ROS production pathways [31,32]. The importance of Ang2 in DKD pathogenesis is further evidenced through the renoprotective benefits associated with Angiotensin Converting Enzyme inhibitors (ACEi) and Angiotensin Receptor Blockers (ARB), medication classes that limit the effects of Ang2 [30].

### 2.3. Inflammatory and Immune Mechanisms

The main cytokines implicated in DKD pathogenesis are the inflammatory cytokines Interleukin-1 (IL-1), Interleukin-6 (IL-6), Interleukin-18 (IL-18) and TNF-α, and the pro-fibrotic cytokine TGF-β [36,37]. In DKD these cytokines act within the kidney to promote endothelial permeability, inflammation and glomerular structural changes [36]. Activation of the pro-inflammatory complement cascade is observed in DKD [38] and blocking complement component C5a has been shown to attenuate renal injury in diabetes [39]. Greater numbers of macrophages and T lymphocytes are observed in the kidneys of DKD patients [40], which are activated by hyperglycaemia and advanced glycation endproducts (AGEs) [36,41,42]. In DKD, macrophages promote ROS, cytokine production and renal vascular inflammation [36]. Further, DKD is associated with lowered circulating levels of regulatory T cells (Tregs) [43], which are mediators of the immune response. Depleted Treg populations enhance inflammation [44] and circulating Treg levels are negatively correlated with urine albumin to creatine ratio (uACR) [45]. Interleukin-10 (IL-10), an important promoter of Tregs, is also downregulated in DKD [46]. The transcription factor NF-κB, a key regulator of innate immunity, also promotes DKD pathogenesis by furthering oxidative stress and renal inflammation [19,36]. In DKD the activity of NF-κB is enhanced by hyperglycaemia and AGEs [23,36]. This stimulates the production of cytokines TGF-β, IL-1, IL-6 and TNF-α, the mitochondrial generation of ROS, and the RAAS [23,24].

### 2.4. Oxidative Stress

Oxidative stress is an important feature of DKD [24], and is characterised by the unopposed action of ROS due to an imbalance between ROS and antioxidant production [21]. In DKD this occurs through the pathological upregulation of ROS production pathways [21,24,47] which is primarily promoted by Ang2, hyperglycaemia, AGEs, and proinflammatory cytokines [24]. DKD is also associated with decreased concentrations of the antioxidants superoxide dismutase and glutathione [23,48], which normally act to neutralise the effects of ROS [7]. Within the glomerulus, oxidative stress leads to podocyte and endothelial dysfunction, expansion of the mesangial matrix and increased production of TGF-β [24]. It also enhances the renal influx of inflammatory cells and cytokines and promotes tubulointerstitial fibrosis [24]. Further, oxidative stress consumes nitric oxide, which diminishes the dilatory capacity of blood vessels, increasing their exposure to direct stress.

### 2.5. Advanced Glycation Endproducts

AGEs are post-translational modifications of proteins produced through the Maillard reaction [21]. Within the body, their formation is promoted by hyperglycaemia, oxidative stress and dyslipidaemia [19,21,23], however, they can also be consumed through the diet. Foods containing high levels of AGEs include processed foods and red meat [49]. In part, AGEs exert their effects through binding to the receptor for AGEs (RAGE) which is overexpressed in the diabetic kidney [50]. Binding to RAGE upregulates NF-κB [19] and promotes oxidative stress and renal inflammation [51]. It has also been demonstrated that increased dietary AGE consumption, is positively associated with albuminuria and intestinal permeability in a db/db mouse model of T2DM [52].

## 3. The Gut-Kidney Axis

The gut-kidney axis is an an emerging pathway in DKD pathogenesis that describes the bidirectional dynamic interaction that exists between the gastrointestinal system, in particular the large intestine, and the kidneys [42,53]. This is a newly implicated mechanism in DKD pathogenesis and is thought to contribute towards disease development and progression via two main mechanisms. The first relates to observed alterations in the composition and functionality of the gut microbiome in DKD patients [54], a state often referred to as gut dysbiosis [42,43]. The second relates to the growing evidence base which suggests that DKD is associated with impaired intestinal barrier function [52] (Figure 3). Whilst beyond the scope of this review, it is worthwhile to note that there is emerging evidence that commonly used anti-hyperglycaemic agents prescribed in diabetes, including metformin, sulfonylureas and GLP-1 receptor agonists, may alter the microbiome, as recently reviewed in detail by Cao et al. [55].

### 3.1. The Gut Microbiome and DKD

The gut microbiome describes the collective genetic information of the trillions of different bacteria, viruses, protozoa, and fungi that colonise the digestive tract [42,43,56]. In humans, the gut microbiome is primarily comprised of bacteria belonging to the *Firmicutes*, *Bacteroidetes*, *Actinobacteria* and *Proteobacteria* phyla [57]. Under normal conditions this complex network exists mutualistically with its host [58] and is highly responsive to changes within the intestinal environment [59]. Its composition is most significantly determined by diet [60], however, hormonal, genetic and environmental factors are also influential [42]. DKD is associated with significant alterations in the composition and functionality of the gut microbiome [16,42,43,54] which is believed to contribute to the chronic sub-acute inflammatory state that is characteristic of this condition. These changes occur due to shifts within the intestinal environment that place selection pressures on healthy microbial communities [61]. The overall themes in the gut dysbiosis associated with DKD are an increase in bacteria capable of producing uremic toxins, and a reduction in bacteria that promote the production of short chain fatty acids (SCFA) [62].

As compared to healthy individuals, DKD is associated with reduced abundance of *Prevotella*, *Lactobacillus* and *Bifidobacterium* species [41,60,63], all of which promote SCFA production [41]. SCFAs are molecules that may be protective in DKD as they promote intestinal health and assist in downregulating inflammation [43,61]. These bacterial populations are thought to decrease as a result of being outcompeted by uremic-toxin producing bacteria [61] which are favoured by the alkaline intestinal environment observed in DKD [60]. This increase in intestinal pH occurs due to a rise in intestinal concentrations of uremic toxins and a decrease in SCFAs [61,64]. A reduction in SCFA-producing bacteria is also thought to be influenced by dietary factors [60]. In later stages of DKD progression, patients are advised to limit their consumption of the electrolytes potassium and phosphorous [65] which can be found in fibre rich foods such as fruit, vegetables and wholegrains. Therefore, limiting these nutrients may lead to a diet deficient in fibre [41], which is indicated by the average fibre intake of CKD patients being significantly below the daily recommended targets [65,66]. Low dietary fibre intake is associated with a decrease in SCFA-producing bacteria, as fibre is their main fuel source [66].

As kidney function declines, so too does its ability to eliminate urea and other uremic toxins. Therefore, these substances gradually accumulate within the bloodstream and gastrointestinal tract [60,62]. Urea is a by-product of normal protein metabolism and uremic toxins are created through the fermentation of protein within the large intestine by proteolytic bacteria. Uremic toxins of note include *p*-cresol sulfate, phenyl sulfate, ammonia and indoxyl sulfate [46]. Increased concentration of urea and uremic toxins promote the growth of uremic toxin producing bacteria [41]. This shift was evidenced by Wong et al. who observed that patients with ESKD had greater proportions of urease positive bacteria, which metabolise urea to form ammonia, alongside increased proportions of bacteria capable of producing indoxyl sulfate and *p*-cresol sulfate [41]. The presence of uremic toxins also promotes the progression of DKD [16,43,67]. In particular, ammonia is associated with increased intestinal pH and intestinal barrier permeability [46], and *p*-cresol sulfate levels are positively correlated with albuminuria risk and in vitro cause podocyte injury, renal fibrosis and inflammation in DKD [67].

Additional microbial changes were observed by Vaziri et al. who compared the stool samples of 12 healthy human controls with 24 ESKD patients, 15 of whom had underlying DKD. In those with ESKD increases in the *Actinobacteria*, *Firmicutes*, and *Proteobacteria* phyla were observed [60]. The greatest increases occurred in the *Pseudomonas* and *Enterobacteriaceae* species, members of the *Proteobacteria* phylum [60]. This phylum contains Gram-negative bacteria linked with elevating systemic concentrations of Lipopolysaccharide (LPS) which can contribute to metabolic endotoxemia and inflammation [63,68]. Increased levels of the *Proteobacteria* phylum were also observed in T2DM patients with Stage 2 DKD, as compared to T2DM patients without DKD, and healthy controls by Tao et al. [63]. Tao et al. also demonstrated that patients with DKD and T2DM were able to be distinguished from T2DM patients without DKD through the presence of increased levels of *Escherichia shigella*, a member of the *Proteobacteria* phylum. This bacterium promotes disruption of the intestinal barrier [63] and is positively correlated with DKD progression and levels of indoxyl sulfate [63,69].

### 3.2. Intestinal Barrier Disruption and DKD

The intestinal barrier acts as both a chemical and physical barrier to separate the contents of the colon from that of the bloodstream [66]. It is maintained through epithelial tight junctions with an overlying mucous layer composed of mucins [16]. In DKD it is hypothesised that the intestinal barrier is compromised [18]. There are several factors which suggest this mechanism to exist in DKD. Firstly, intestinal barrier disruption has been observed in T2DM and CKD patients [62] as well as in mouse models of DKD [52,70]. Additionally, the gut microbiome changes observed in DKD are associated with a decline in intestinal barrier integrity [16]. A decrease in SCFA-producing bacteria leads to decreased butyrate production, an SCFA which promotes the intestinal barrier [43], and increased urease positive bacteria promote ammonia production which degrades the intestinal epithelium [66]. Additionally, urea is directly toxic to the intestinal barrier as it weakens epithelial tight junctions [60,71].

An impaired intestinal barrier may contribute to the subacute inflammatory state observed in DKD through allowing microbes, toxins, and bacterial by-products to enter the bloodstream [18,72]. An example of this is LPS, a cell wall component of Gram-negative bacteria [46]. Translocation of LPS across the intestinal barrier triggers the deregulation of both the adaptive and innate immune responses and promotes the production of IL-6, IL-1 and TNF-α [16,46]. Increased levels of LPS within the bloodstream also causes metabolic endotoxemia, a proinflammatory state [46]. Given the role of the gut-kidney axis in DKD pathogenesis, and the ability for diet to influence gut microbiome composition, there is growing interest to explore the use of dietary intervention as a therapeutic and preventative tool in DKD. Diets already proven to be of some benefit in DKD include the DASH diet [73], the Mediterranean Diet [74] and a vegetarian diet [75], all of which promote the consumption of fibre and discourage the intake of red meat, salt and processed foods.

## 4. Dietary Fibre

Dietary fibre, often thought of as a single entity, in fact refers to a family of edible plant-based carbohydrate polymers that are resistant to digestion by endogenous enzymes within the human gastrointestinal tract [76,77,78] (Figure 4). Whilst the exact definition of dietary fibre can slightly differ between organisations, it is generally accepted that dietary fibre encompasses non-starch polysaccharides, resistant oligosaccharides, and resistant starch [77].

The unique structure of each fibre type dictates its physiochemical properties and functional effects [79]. When considering function, dietary fibres can be broadly grouped into three main functional classes; bulking fibres, viscous fibres, and fermentable fibres [79,80]. Bulking fibres increase stool bulk which aids in stimulating intestinal peristalsis and shortening intestinal transit time to promote stool regularity [81]. In contrast, viscous fibres combine with water to form a gel which prolongs intestinal transit time whilst also slowing the absorption of nutrients across the intestinal lumen [79,80]. Fermentable fibres can be fermented by specific commensal bacteria of the colon which promotes the maintenance of a healthy intestinal environment and microbiome community. Each fermentable fibre is associated with differing metabolic responses and physiological effect, as the location, speed and type of bacteria involved in the fermentation process varies between fibre type [79,82]. Whilst most fibres tend to be associated with one functional class, they often exert more than one effect, which may be beyond the three classes described. However, given that no one fibre possess all beneficial effects, eating a large range of fibres from all functional classes is essential to optimise health outcomes [80]. Harnessing the unique functional capabilities of individual fibre types for therapeutic purposes is a growing area of interest, particularly in gastrointestinal conditions [79]. Given the role of the gut-kidney axis in DKD, a similar approach utilising fermentable fibres may be beneficial.

### 4.1. Resistant Starch

RS, like digestible starch, is composed of glucose monomers bound by α-glycosidic bonds in the form of amylose and amylopectin polymers [83]. However, unlike digestible starches, due to physical and/or chemical properties, these bonds within RS are inaccessible to the digestive enzymes of the small intestines [84]. Thus, RS arrives undigested within the large intestine where it can be fermented by specific bacteria to produce SCFAs. As such, RS is considered a type of fermentable dietary fibre. There are five RS subtypes and differentiating between these is important as each exerts different effects [85] (Table 1).

Whilst there are no official guidelines in regard to RS consumption [91], it is generally agreed that a daily intake of at least 15–20 g of RS is required to observe health benefits [92]. However, available data suggests that RS consumption in the United States [92], Australia [93] and Europe [94] sits between 3–9 g, well below this suggested amount. This low figure reflects low dietary fibre consumption across western countries [95] as well as a lack of public knowledge about RS, its food sources and the importance of a varied dietary fibre intake. Additionally, estimating RS intake is made difficult as there is a lack of dietary assessment tools and food databases that quantify its consumption, and the amount of RS in a particular food product is highly dependent on the way it is prepared [93]. Foods that have been reported to be high in RS include lentils (3.4 g RS/100 g), muesli (3.3 g RS/100 g), chickpeas (2.6 g RS/100 g), kidney beans (2.0 g RS/100 g) and buckwheat (1.8 g RS/100 g) [96].

#### 4.1.1. Resistant Starch and the Gut Microbiota

RS is a central fuel source for saccharolytic bacteria, a term referring to the ability of these bacteria to digest and ferment carbohydrates. Therefore, RS can positively influence the composition of the gut microbiome by promoting and sustaining these bacterial populations [17,97]. The majority of saccharolytic bacteria belong to either the *Ruminococcus*, *Bifidobacterium*, *Lactobacillus* or *Eubacterium* species [98]. Of these, it is well established that RS increases populations of *Bifidobacterium* and *Lactobacillus* [46,99,100,101] which are both observed to be decreased in DKD [41,60,63]. In healthy human participants, a double-blind cross over study by Martinez et al. compared the individual effects of RS2 and RS4 with a native starch control [86]. Supplementation with 33 g/day of RS2 for three weeks showed increased levels of *Bifidobacterium Adolescentis*, *Eubacterium Rectale* and *Ruminococcus Bromii* [86]. These bacteria are involved in the digestion of RS to create butyrate, an SCFA, (Figure 5) and their increase in response to RS2 is well documented [102,103]. In a population of elderly patients RS2 has also been demonstrated to decrease *Proteobacter**ia* phyla concentrations [104], which are likely elevated in DKD [60,63]. Regarding RS4, Martinez et al. demonstrated that 30 g/day for three weeks was associated with an increase in *Parabacteroides distasonis* and *Bifidobacterium* species whereas *R. Bromii* and *Eubacterium* species decreased [86]. Interestingly, whilst both RS2 and RS4 increased *Bifidobacterium* species, RS2 achieved this change at a slower rate [86]. In obese males, a three-week intervention with 25.5 g/day of RS3 was not associated with changes to *Bifidobacterium* populations, but *E. rectale* and *Ruminococcus* species increased [105]. These findings highlight the differing effects of each RS subtype.

#### 4.1.2. Resistant Starch and Short Chain Fatty Acids

SCFAs are molecules formed through the fermentation of dietary fibre by saccharolytic bacteria [46,76]. Within the large intestine, the main SCFAs produced are butyrate, acetate, and propionate [17]. Butyrate exerts local effects whereas acetate and propionate are absorbed via the portal vein to act systemically [43]. In both animals and humans, RS supplementation is associated with enhancing SCFA production [87,102,103,106,107]. Of particular interest is its ability to increase butyrate concentrations, as this SCFA promotes intestinal health [46] as it is the preferred fuel source for colonocytes [76]. Butyrate plays a key role in maintaining functional intestinal barrier integrity, which is thought to be compromised in DKD [43]. It achieves this through stimulating the production of intectin, tight junction proteins, and GLP-2 [46], which strengthens enterocyte structure and intercellular junctions [64]. Additionally, butyrate upregulates mucin production to maintain the thick mucus layer overlying enterocytes [108]. Enhancement of the intestinal barrier prevents the translocation of substances such as LPS, thus preventing the development of metabolic endotoxemia [17,64].

Regarding inflammation, in mouse models, butyrate has been observed to suppress levels of the proinflammatory cytokine IL-6 [17], and increase both circulating Treg populations and levels of IL-10 [109,110]. Further, in ulcerative colitis patients, butyrate supplementation reduced activity of NF-κB [111] and in T2DM patients, it increased levels of GLP-1 [112]. GLP-1 reduces oxidative stress [113], improves glucose and insulin regulation [114] and has been shown to decrease intraglomerular pressure [114]. Interestingly, GLP-1 receptor agonists have been associated with gut microbiota changes, notably an expansion of *Akkermansia muciniphila* [55]. Butyrate also limits oxidative stress by increasing levels of the antioxidant glutathione in healthy humans [115]. The anti-inflammatory effects of butyrate are primarily mediated through its binding to the GPR43 and GPR109A receptors [17,110], and its ability to inhibit histone deacetylases, which are found to be dysregulated in DKD [116]. Their inhibition in DKD is associated with the downregulation of TGF-β [117]. Lastly, butyrate lowers large intestinal pH [97,118,119] which protects against the accumulation of uremic toxin-producing bacteria which prefer a high pH environment [41]. However, a recent intervention trial in patients with T1DM providing 3.6 g/day sodium butyrate for 12 weeks found no changes in HbA1c, UACR or fecal calprotectin (a marker of intestinal inflammation) [120]. A similar dose of butyrate (4 g/day) for 6 months in ulcerative colitis patients did reduce fecal calprotectin levels [121], though the baseline calprotectin levels were over four-fold higher in this study compared with the study in T1DM, suggesting that butyrate may be beneficial only in severe intestinal inflammation. Given that butyrate alone does not improve renal outcomes in diabetes, this may indicate that interventions that alter the gut microbiota consortium are required, rather than the use of microbial metabolites.

#### 4.1.3. Resistant Starch, Inflammation and Oxidative Stress

Three recent meta-analyses have assessed the impacts of RS on inflammation and oxidative stress, however each published differing results and only assessed a limited number of inflammatory markers. In a meta-analysis of 16 papers, Wei et al. observed RS to increase circulating total antioxidant capacity (TAC) and decrease TNF-α and IL-6 [122]. RS was also demonstrated to increase TAC in a meta-analysis of 13 papers by Lu et al. in addition to an overall reduction to CRP [123]. Lu et al. did not find RS to reduce levels of IL-6 or TNF-α, however, one study appeared to greatly influence this result, and if excluded, RS was associated with a decrease in both inflammatory markers [123]. Lastly, in a meta-analysis of eight studies exclusively looking at the effects of RS2, RS2 was found to have no overall impact on IL-6, TNF-α or CRP [124]. These unclear and inconsistent results highlight a key issue relating to all RS studies, being that there is a significant lack of comparable study designs assessing the same clinical endpoints. It is rare that studies investigating RS share similar patient populations, duration of intervention or type of control. Further, despite most studies utilising RS2, the amount, and method of delivery of RS are rarely the same. This would make the direct comparison of studies within each meta-analysis challenging and questions the relevancy of their results.

#### 4.1.4. Resistant Starch and Glucose Control

The exact effect of RS on glucose control remains contentious as many studies report conflicting results. Contributing to this is the lack of consistency in study design which makes the direct comparison of findings challenging. The most established effect of RS on glucose control is its ability to lower postprandial blood glucose concentrations in DM patients [46,125,126,127]. However, this effect does not appear to translate to those with prediabetes [128]. In regard to FPG and HbA1c the effects of RS are less established. A number of studies have demonstrated no effect of RS on FPG concentrations [126,128,129,130,131]. However, in a trial by Meng et al. investigating RS2 intervention in T2DM patients with DKD, RS supplementation was associated with lowered FPG levels [7]. These findings were partially corroborated by Kwak et al. who observed lower FPG in their RS intervention cohort, although these changes did not achieve statistical significance [127]. In relation to HbA1c, RS supplementation did not appear to affect this measure in patients with well-controlled DM [126,129]. Interestingly, in studies utilising patient populations with higher baseline HbA1c levels, RS supplementation decreased HbA1c [7,131]. The impacts of RS on blood glucose levels are mainly attributed to its low GI, slow rate of digestion and low energy density as compared to other carbohydrates [7,46,74]. This is likely the explanation behind its effects on postprandial glucose control. However, there is evidence to suggest that RS may alter the expression of certain genes related to glucose homeostasis [132], although these effects have not been observed in humans.

The effects of RS across all measures of glucose control appear to be related to pre-existing diabetic control. From the studies outlined above it appears those with higher baseline HbA1c and more established T2DM are more responsive to RS supplementation. The majority of RS intervention studies have been conducted in T2DM populations. In patients with T1DM, RS has been shown to reduce single meal postprandial glycaemia [133], in a similar manner as was seen in T2DM [134], suggesting that the beneficial effects on reducing hyperglycaemia may be comparable between T1DM and T2DM. However, a recent 6-week pilot study with an acetylated, butylated RS did not lead to improvements in glucose control or insulin requirements in T1DM, despite altering the microbiome and increasing fecal SCFA levels [135]. Interestingly, the effects of RS on blood glucose measures appeared unrelated to the amount or duration of RS supplementation. Regardless of the exact relationship between RS and glucose control, consumption of low GI carbohydrates in replacement of high GI carbohydrates should always be encouraged given their effects on appetite and weight control [136].

#### 4.1.5. Factors Influencing the Effects of Resistant Starch

Whilst RS consumption is associated with several beneficial effects (Figure 6), the degree to which these effects are observed appears to be influenced by several factors. Firstly, RS appears to display a dose–response relationship [17,46,119], with increased consumption being associated with greater effect. It is widely considered that at least 20 g of RS is required to observe a positive effect on cardiometabolic markers [7,76], however, supplementation with as little as 1.74 g/day was noted to be impactful on the gut microbiome [137]. The type and source of RS also impacts its effect. In particular, different food sources of RS2 are associated with different microbial and SCFA responses [102,107]. In addition to these practical factors, there is significant variation to individual response to RS supplementation [87,101,103,105]. This variation in individual response likely reflects differences in baseline gut microbiome composition and co-existing dietary habits [101,102,103,105]. Additionally, genetic and health related factors are also important. For example, overweight individuals may require a greater amount of RS to exert an effect [76], as may African American populations [138]. This information suggests that RS may be of greatest benefit if utilised as a personalised therapy [87,102,103], as the type, amount and source of RS required to achieve a particular effect likely differs between individuals depending on their baseline health status and gut microbiome.

### 4.2. Resistant Starch and Diabetic Kidney Disease—Animal Models

There is a significant overlap between the effects of RS and the mechanisms involved in DKD pathogenesis, especially in relation to the inflammatory and gut-kidney axis pathways. Therefore, it appears likely that RS has the potential to promote kidney health in DKD. Currently, limited evidence exists examining the relationship between RS and DKD, especially in early stages of disease.

#### 4.2.1. Resistant Starch in T2DM Models

Koh et al. examined the use of RS in Diabetic Fatty Rats, a model of T2DM, where rats were fed a diet containing either corn starch or RS2 for six-weeks [139]. This study observed RS supplementation to decrease albuminuria, significantly lower blood glucose concentrations and limit renal histopathological damage [139]. In fact, levels of albuminuria in RS fed rats were no different to healthy controls [139]. Similarly, we have observed that in the db/db mouse model of T2DM, RS2 supplementation was protective against albuminuria induced by high dietary AGE consumption [52].

#### 4.2.2. Resistant Starch in T1DM Models

In T1DM mice induced using streptozotocin (STZ), Li et al. concluded that 12-weeks of 62.3% RS2 supplementation led to an increase in *Bifidobacterium* populations, an increase in systemic and fecal SCFAs, a decrease in UACR and protection against glomerular hypertrophy and podocyte loss [17]. Further, Li et al. demonstrated that RS2 lowered levels of macrophages within the renal interstitium and decreased the expression of IL-6, TNF-α and TGF-β [17]. However, these findings are not consistent with other studies investigating the use of RS in STZ-induced diabetes animal models. In a four-week trial by Koh et al., neither 5%, 10% or 20% RS2 diets were shown to affect UACR, IL-6 or TNF-α [140]. Additionally, we observed no change to kidney function or intestinal permeability after 24 weeks of 12.5% RS2 intervention [53]. These conflicting results may reflect Li et al.’s use of a markedly higher concentration of RS2 supplementation than the latter two studies. Whilst this may have increased the ability of RS2 to be effective in showing renoprotective and inflammatory benefits, this supraphysiological dosing would not be practical for long term human consumption.

Another factor worth considering is the timing of RS supplementation in relation to T1DM induction with STZ. STZ possesses antibiotic-like properties [141], and as such alters and depletes the gut microbiome. However, these effects appear temporary and resolve over time [17]. Both Koh et al. and Snelson et al. began RS very shortly after STZ induction and this timing may have negatively impacted the ability of RS to impact inflammatory or renal markers [53,140]. In contrast, Li et al. only commenced RS supplementation three weeks after STZ induction [17]. This timing allows greater recovery of the gut microbiome which would facilitate a more favourable environment for RS action.

Lastly, the timing of RS intervention in relation to the onset of DM may also play a role in its effectiveness. Smazal et al. observed that RS2 given 21 days before STZ induction, attenuated histopathological renal damage and was protective against albuminuria development in mice [142]. Therefore, RS may exert the greatest effects when used as a preventative tool. Interestingly, across all four studies investigating the use of RS in T1DM mice, no studies observed RS to impact any blood glucose measures [17,53,140,142]. This suggests that the effects of RS in DKD are unrelated to hyperglycaemic control.

#### 4.2.3. Resistant Starch in CKD Models

Whilst not specific to DKD, RS2 supplementation is associated with improvements in biochemical and histopathological renal outcomes in mice with both 5/6 nephrectomy and adenine-induced CKD [143,144,145]. Specifically, in adenine-induced CKD mice, RS2 reduced levels of oxidative stress and uremic toxins [143,145] and increased populations of *R. Bromii* and *Bifidobacterium* species [143]. An overview of animal studies utilising RS in CKD and DKD is outlined in Table 2.

### 4.3. Resistant Starch and Diabetic Kidney Disease—Clinical Trials

#### 4.3.1. Resistant Starch in Early Stage DKD

A study by Meng et al. used 17.41 g of RS in a population of Stage 2 DKD patients with T2DM over 12 weeks, the only trial at this time investigating the impacts of RS in early stage DKD [7]. Meng et al. concluded that RS supplementation led to a decline in serum uric acid, increased antioxidant concentrations and an improvement in both lipid and glucose profiles [7]. However, no significant differences in UACR, Blood Urea Nitrogen, TNF-α or IL-6 were observed within the intervention cohort [7]. This study did not report on dietary intake of participants, so it is unknown whether differences in habitual RS intake may act as a confounding factor for the intervention.

#### 4.3.2. Resistant Starch in End Stage Kidney Disease

RS supplementation has been most extensively investigated in ESKD patients undergoing haemodialysis. In this patient group, RS2 has been associated with decreased Blood Urea Nitrogen, IL-6, TNF-α and serum creatinine [44,146,147] as well as reduced levels of oxidative stress and constipation [44]. A meta-analysis analysing five RCTs investigating the effects of RS2 on ESKD patients undergoing haemodialysis found that RS supplementation did not impact levels of *p*-cresol sulfate or indoxyl sulfate [146]. As these uremic toxins are both promoters and indicators of altered microbial consortium, it may suggest that RS supplementation is less effective in altering the microbiome composition in later CKD stages. This hypothesis is supported by a study which investigated the impact of RS2 on five species of SCFA producing bacteria: *Faecalibacterium*, *Parabacteroides*, *Bifidobacteria*, *Ruminococcus* and *Prevotella*, in ESKD haemodialysis patients. Of these, a significant increase was only observed in the *Faecalibacterium* species [147]. Whilst these studies ranged in duration from 4–8 weeks, dietary interventions can alter the microbiome within 24 h [148], thus the duration of these trials would be sufficient to alter the gut microbiota. Therefore, it could be possible that for RS to have the greatest impacts within the gut microbiome, it needs to be introduced in earlier disease stages. An overview of the use of RS in patients with CKD and DKD is outlined in Table 3.

## 5. Future Directions

Much research has been conducted over recent decades exploring the mechanisms by which resistant starch exerts beneficial effects, future research would benefit from the development of RS food databases. Whilst current food databases provide data on dietary fibre as a whole, there is a dearth of data about RS content in many foods. The development of food databases with RS content would permit an assessment of habitual resistant starch intake which is of relevance for interventional studies, where RS supplementation may lead to a decrease in habitual total fibre intake [153]. The development and dissemination of these databases would also permit large scale epidemiological studies of RS intake, which could complement the intervention trials reported here.

## 6. Conclusions

Utilising RS in early stage DKD patients may present a cost effective, non-medication-based tool, to assist in attenuating disease development and progression. RS has the potential to affect a number of mechanisms implicated in DKD progression through its capacity to increase SCFA production, improve intestinal barrier integrity, downregulate inflammatory pathways and restore a healthy gut microbiome. Despite these promising links, there remains a lack of long term, comparable studies exploring the use of RS in DKD. There are also no current studies investigating habitual dietary RS intake in relation to DKD outcomes. Exploring this baseline relationship is important, as not only will it add to a currently limited evidence base, but it will also provide greater context to guide future clinical trials and may highlight areas for further clinical investigation. Additionally, this information may assist in guiding the use of RS in clinical settings if a positive association is established. Therefore, investigating the use of RS as an adjunct therapy for DKD is important as it may provide a novel treatment strategy to combat this significant health issue.

## Figures and Tables

**Figure 1 nutrients-14-04547-f001:**
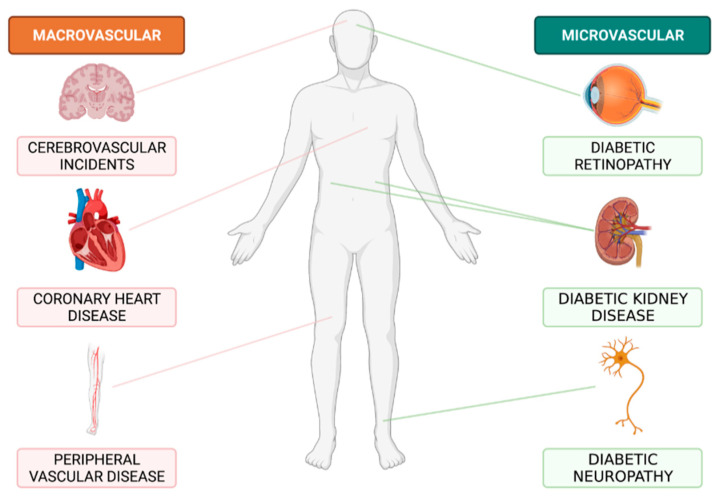
Long-term major complications of diabetes mellitus. Created with Biorender.com.

**Figure 2 nutrients-14-04547-f002:**
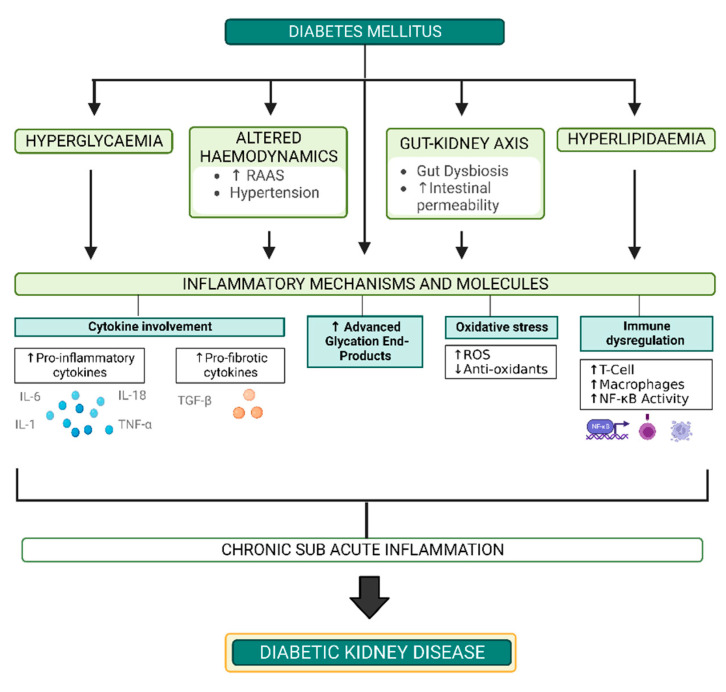
Overview of the mechanisms implicated in DKD pathogenesis. RAAS: Renin Angiotensin Aldosterone System, ROS: Reactive Oxygen Species, NF-κB: Nuclear Factor Kappa B, IL-6: Interleukin 6, IL-1: Interleukin 1, IL-18: Interleukin 18, TNF-α: Tumour Necrosis Factor Alpha, TGF-β: Transforming Growth Factor Beta, ↑: Increased, ↓: Decreased. Created with Biorender.com.

**Figure 3 nutrients-14-04547-f003:**
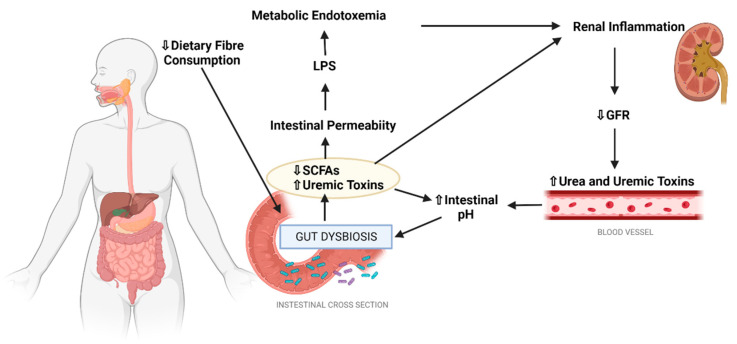
Overview of the gut-kidney axis in DKD. GFR: Glomerular Filtration Rate, LPS: Lipopolysaccharide, SCFA: Short Chain Fatty Acid, ↑: Increased, ↓: Decreased. Created with Biorender.com.

**Figure 4 nutrients-14-04547-f004:**
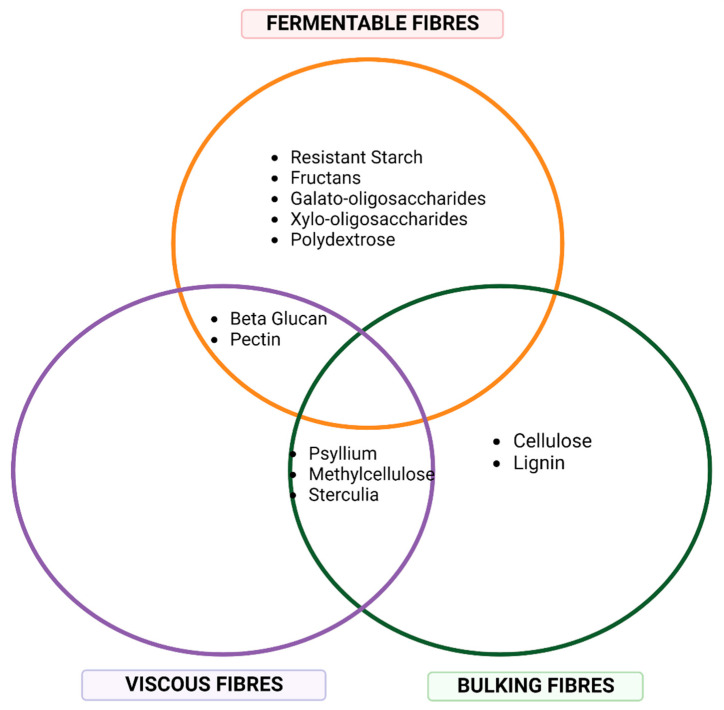
A venn diagram placing dietary fibre types in relation to their associations with bulking, fermenting and viscous functions. This is not a comprehensive overview of fibre types or functional associations but aims to demonstrates the wide diversity of the dietary fibre class. Created with Biorender.com.

**Figure 5 nutrients-14-04547-f005:**
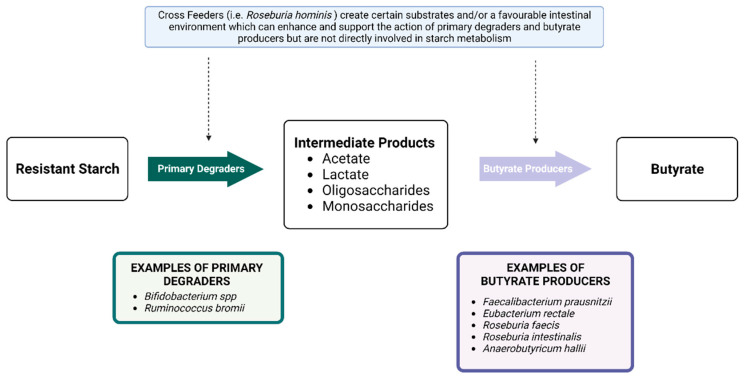
An overview of bacteria associated with resistant starch and the production of butyrate. Created with Biorender.com.

**Figure 6 nutrients-14-04547-f006:**
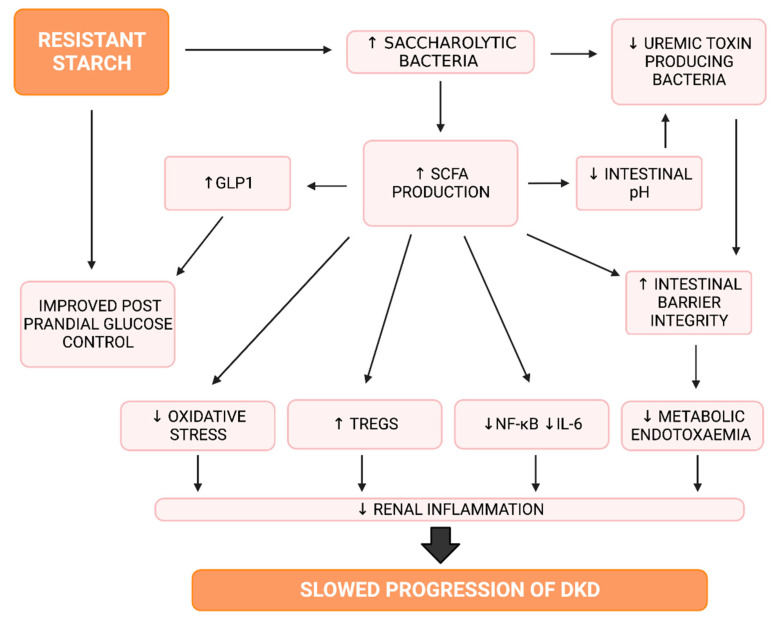
An overview of the mechanisms of resistant starch. GLP1: Glucagon Like Peptide 1, IL-6: Interleukin 6, NF-κB: Nuclear Factor Kappa B, SCFA: Short Chain Fatty Acid, Treg: Regulatory T cells, ↑: Increased, ↓: Decreased. Created with Biorender.com.

**Table 1 nutrients-14-04547-t001:** Types of Resistant Starch.

Type	Description	Food Sources
RS1	RS1 refers to starch molecules encased within an intact plant cell wall. These starch molecules are physically inaccessible to the digestive enzymes of the upper alimentary tract, as humans lack the ability to digested plant cell wall components [82,86]. Therefore, the resistance of RS1 can be lost through any process which damages this protective cell wall barrier, such as milling, grinding or mastication [87].	Legumes, Seeds, Wholegrains
RS2	RS2 refers to tightly organised ungelatinized starch granules [46]. Whilst the exact mechanisms of its resistance are not fully understood [82], it is thought that its dense structure makes it difficult for digestive enzymes to effectively access and attach to these starch molecules [82,88].	Raw potato, Unripe bananas, High Amylose Maize Starch (HAMS) [84]
RS3	RS3 is created through the process of retrogradation [76]. To undergo retrogradation, starch first needs to undergo gelatinization which occurs when starch is heated and becomes more viscous as water molecules enter the starch granule [46]. As the starch cools down, retrogradation then occurs, where its structure reforms to create a more tightly packed, inaccessible crystalline structure [88].	Heated and cooled potatoes, rice
RS4	RS4 is created by the chemical modification of starch molecules. Such processes include dextrinization, substitution of functional groups and esterification [43,82]. RS4 encompasses a large range of different molecules given the various combinations of starch bases and chemical processes that are available [89].	
RS5	RS5 has traditionally referred to starch-lipid complexes, created through the combination of long side chains of amylopectin or amylose with lipids or free fatty acids. This structure limits accessibility to digestive enzymes [82] and can be both naturally or artificially derived [88]. More recently, more resistant starch complexes have been identified such as starch-protein complexes and starch-glycerol complexes [90].	

**Table 2 nutrients-14-04547-t002:** Overview of Studies Investigating Resistant Starch in Animal Models of Chronic Kidney Disease and Diabetic Kidney Disease.

Study	Population	Intervention	Control	Group Size (*n*)	Duration (Weeks)	Alb	Ucr	CrCl	BUN	Renal Histology	Inflammatory Markers	Intestinal Markers
Preclinical CKD Models												
[145] ^	Male Sprague Dawley rats adenine-induced CKD	59% HAMS	Amylopectin low fibre diet	9	3	--	--	↑	--	↓ tubulointerstitial injury	↓ TGF-β, ROS, MCP-1	Δ microbiome
[143] ^	Male Sprague Dawley rats adenine-induced CKD	59% HAMS	Amylopectin low fibre diet	9	3	--	--	↑	↓	--	--	↓ pH Δ microbiome
[144]	Male C57BL6 mice 5/6 nephrectomy	59% HAMS	Regular control diet	4	4	--	--	--	↔	↓tubulointerstitial injury	--	Δ microbiome
Preclinical Diabetes Models												
[142]	STZ treated Male Sprague Dawley rats	55% HAMS (20% RS)	55% Corn Starch	5	5	↓	--	--	--	↓ proximal tubular injury	--	--
[139]	Male Zucker Diabetic Fatty Rats	55% HAMS (35% RS)	Corn Starch control diet	8	6	↓	↑	--	--	--	--	--
[140]	STZ-treated male Sprague Dawley rats	13.75%, 27.5% or 55% HAMS (5%,10% or 20% RS)	Corn Starch control diet	8	6	↔	↔	--	--	--	--	--
[53]	STZ-treated male Gpr109a^−/−^ mice	25% HAMS (12.5% RS)	20% starch + 5% cellulose	10–11	24	↔	--	--	--	↔ renal hypertrophy ↔ Glomerulosclerosis Index	↔ MCP-1	--
[17]	STZ-treated male Gpr109a^−/−^ mice	63.6% RS (source not outlined)	Normal Chow, Zero Fibre	5–10	12	↓	--	--	--	↓ Glomerular hypertrophy ↓ Podocyte injury ↓ Interstitial fibrosis	↓ TNF-α ↓TGF-β ↓ IL6	↑ SCFA
[52]	Male db/db mice	25% HAMS (12.5% RS)	20% starch + 5% cellulose	12	10	↓	--	↔	--	--	--	↓ in vivo gut permeability

^: same cohort, --: not assessed, ↔: no change, ↓: decreased, ↑: increased, Δ: changed. Alb: albuminuria, Ucr: Urinary Creatinine, CrCl: Creatinine clearance, BUN: Blood urea nitrogen, TNF-α: Tumour necrosis factor alpha, TGF-β: transforming growth factor beta, IL-6: interleukin 6, MCP-1: Monocyte chemoattractant protein-1, ROS: Reactive Oxygen Species.

**Table 3 nutrients-14-04547-t003:** Overview of Studies Investigating Resistant Starch in Humans With Chronic Kidney Disease And Diabetic Kidney Disease.

Study	Population	Intervention	Control	Group Size (*n*)	Duration (Weeks)	Alb	SCr	BUN	Uremic Toxins	Inflammatory Markers	Microbiota
Chronic Kidney Disease											
[149]	Stable haemodialysis	15 g/d HAMS (60% RS)	15 g/d waxy corn starch	20	6	--	--	↔	↓IS ↔ PS	↔ CRP	--
[150] ^%^	Stable haemodialysis	26 g/d HAMS (16 g/d RS)	20 g/d manioc flour	15–16	4	--	↔	↔	↓IS ↔ PS	↓ IL-6, ↓ TBARS, ↔ hs-CRP	--
[44]	Stable haemodialysis (Diabetic patients excluded)	20 g/d 4 weeks, 25 g/d 4 weeks HAMSRS2 (60% RS)	20 g/d 4 weeks, 25 g/d 4 weeks Wheat flour	22	8	--	↓	↓	--	↓ TNF-α, ↓ IL6, ↓ MDA ↔ hs-CRP, ↔ IL-1β ↔ TAO activity	--
[151] ^$^	Stable haemodialysis (Diabetic patients excluded)	20 g/d 4 weeks, 25 g/d 4 weeks HAMSRS2 (60% RS)	20 g/d 4 weeks, 25 g/d 4 weeks waxy corn starch	21–23	8	--	↓	↔	↓ PC ↔IS	↔ hs-CRP	--
[147] ^$^	Stable haemodialysis (Diabetic patients excluded)	20 g/d 4 weeks, 25 g/d 4 weeks HAMSRS2 (RS% not stated)	20 g/d 4 weeks, 25 g/d 4 weeks waxy corn starch	9–11	8	--	↔	↓		↓ IL6, ↓TNF-α ↓MDA	↑ Faecalibacterium genus ↔ Bifidobacteria genus ↔ Ruminococcus genus ↔ Prevotella genus
[152] ^%^	Stable haemodialysis	16 g/d RS	16 g/day manioc flour	8	4	--	--	--	--	↓ RANTES, ↓ PDGF-BB ↓ IP10, ↔ IL10	--
Diabetic Kidney Disease											
[7]	T2DM with early stage DN aged 18–80	50 g/d high RS flour (17.41 g/d RS)	Control diet (not stated)	37–38	12	↔	↔	↔	--	↔ TNF-α, ↔ IL-6, ↑ SOD	--

^$^,^%^: same cohorts, --: not assessed, ↔: no change, ↓: decreased, ↑: increased, Alb: albuminuria, BUN: Blood urea nitrogen, SOD: Superoxide dismutases, TNF-α: Tumour necrosis factor alpha, TGF- β: transforming growth factor beta, PC: *p*-cresol, IS: indoxyl sulfate, RANTES: Regulated on activation, Normal T Cell Expressed and Secreted (a chemokine), hs-CRP: high sensitivity C-reactive protein, MDA: malondialdehyde, PDGF-BB: Platelet Derived Growth Factor BB, IP-10: Interferon gamma-induced protein 10, IL-10: Interleukin 10, IL-6 = interleukin 6, IL-1β: Interleukin 1 beta, TBARS: Thiobarbituric acid reactive substance, TAO: Total Antioxidant Activity.

## Data Availability

Not applicable.

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
