# Peer review of "Resistant Starch as a Dietary Intervention to Limit the Progression of Diabetic Kidney Disease"

_nutrients, 2022, doi:10.3390/nu14214547_

Round 1
Reviewer 1 Report
Drake et al. summarise in this review the use of resistant starch (RS) as a dietary intrevention to limit the progression of diabetic kidney disease. This review is informative and well written. However, I would suggest some changes:
- please add the results from the study from Tougaard, N.H. et al. (J. Clin. Med. 2022, 11, 3573. https://doi.org/10.3390/ jcm11133573). They show that a "Twelve weeks of butyrate supplementation did not reduce intestinal inflammation in persons with type 1 diabetes, albuminuria and intestinal inflammation" and comment these results. Why do these changes differ from the study reported on patients with type 2 diabetes?
- explain if there is any (expected) difference on the effects of RS in type 1 diabetes compared to type 2 diabetes in preclinical and human models. Since the underlying mechanisms of these two diseases differ largely, it would be important to comment the use of RS dependent on diabetes type.
- when writing in line 325-327 that " Further, in ulcerative colitis patients, butyrate supplementation reduced activity of NF-κB [109] and in T2DM patients, it increased levels of GLP-1", look for studies that have investigated gut microbiom under GLP-1 agonists. Does the use of GLP-agonists lead to changes in gut microbiom that would favour the SCFA production?
- what is known on RS effects and drug interaction with gut microbiome? For more information on bacteria-drug interactions the authors can for example refer to the study from Klünemann et al. 2021 (https://doi.org/10.1038/s41586-021-03891-8). Can the authors speculate on that? Since patients with diabetes are under glucose-lowering medication, some such as Metformin and GLP-agonists known to effect gut microbiom and the gastrointestinal tract, can the potential benefitial effects of RS be modulated from the glucose-lowering medication? What about the SGLT-2 inhibitors? Since they have been shown to excert renal protective effects, do they also effect positively the gut-kidney-axis?
- the authors should comment the duration of study intervention with RS, since the clinical studies listed in the review had a short study of 4-8 weeks, only one study duration. What should be the right duration of RS intake? When should one excpect a change in gut microbiom?
Author Response
Reviewer 1:
Drake et al. summarise in this review the use of resistant starch (RS) as a dietary intrevention to limit the progression of diabetic kidney disease. This review is informative and well written. However, I would suggest some changes:
- please add the results from the study from Tougaard, N.H. et al. (J. Clin. Med. 2022, 11, 3573. https://doi.org/10.3390/ jcm11133573). They show that a "Twelve weeks of butyrate supplementation did not reduce intestinal inflammation in persons with type 1 diabetes, albuminuria and intestinal inflammation" and comment these results. Why do these changes differ from the study reported on patients with type 2 diabetes?
Thank you for bringing this recently published study to our attention. We have included a discussion of this study, and how it relates to other findings in the revised manuscript (lines 352-361).
- explain if there is any (expected) difference on the effects of RS in type 1 diabetes compared to type 2 diabetes in preclinical and human models. Since the underlying mechanisms of these two diseases differ largely, it would be important to comment the use of RS dependent on diabetes type.
We have included discussion of this point in the revised manuscript (lines 405-411).
- when writing in line 325-327 that " Further, in ulcerative colitis patients, butyrate supplementation reduced activity of NF-κB [109] and in T2DM patients, it increased levels of GLP-1", look for studies that have investigated gut microbiom under GLP-1 agonists. Does the use of GLP-agonists lead to changes in gut microbiom that would favour the SCFA production?
Liraglutide has been observed to be associated with increases in Akkermansia muciniphila. We have included reference to this in the revised manuscript (lines 344-346)
- what is known on RS effects and drug interaction with gut microbiome? For more information on bacteria-drug interactions the authors can for example refer to the study from Klünemann et al. 2021 (https://doi.org/10.1038/s41586-021-03891-8). Can the authors speculate on that? Since patients with diabetes are under glucose-lowering medication, some such as Metformin and GLP-agonists known to effect gut microbiom and the gastrointestinal tract, can the potential benefitial effects of RS be modulated from the glucose-lowering medication? What about the SGLT-2 inhibitors? Since they have been shown to excert renal protective effects, do they also effect positively the gut-kidney-axis?
The area of how commonly used anti-diabetic medications affect the microbiome and whether this is a mechanism of action for these pharmaceutical agents is an interesting and quite active area of research. However this is beyond the scope of this review, which is focusing on how resistant starch may be used as a dietary therapy. We note several recent reviews (Merkevičius 2021, PMID: 34684121; Cao 2020, 33071980) have reported on this. The review by Cao 2020, which assessed 64 studies (human and mice) goes into much greater detail than we could here. We have made reference to this in lines 147-150.
Is there a synergistic effect of RS interventions and anti-diabetic medications? No studies have looked specifically at RS, and there is only one published pilot study looking at a dietary fibre mix intervention.
Lee 2019 (PMID: 31237126) conducted an open-label single arm pilot study in 10 T2DM patients who were on metformin and sulfonylurea treatment for at least 6 months. They were given a dietary intervention (consisting of a mixture of 11.7 g/d of plantago seed and 0.39 g/d of ispaghula husk) for 4 weeks. There was no overall effect on insulin sensitivity or glycaemic metrics, though the authors suggest there may have been a responder/non-responder dichotomy, as 3 participants did see slight improvements in glucose AUC following a mixed meal tolerance test (not statistically significant, but likely due to underpowered as a pilot study).
There is also one trial currently recruiting which will test the synergistic effects of metformin with a high mixed fibre supplement (35g/d consisting of 6 g of Oligofructose + 12g of resistant maltodextrin + 12g of acacia gum + 5g of PGX) (Deehan 2021, PMID: 33596993; ClinicalTrials.gov Identifier: NCT04578652)
- the authors should comment the duration of study intervention with RS, since the clinical studies listed in the review had a short study of 4-8 weeks, only one study duration. What should be the right duration of RS intake? When should one excpect a change in gut microbiom?
Dietary interventions can alter the gut microbiome within 24 hours (PMID: 24336217). The timeframes in these studies are sufficient to observe alterations in the microbiome. We have included reference to this in the revised manuscript (lines 526-529)
Reviewer 2 Report
The review article has discussed the mechanisms involved in this pathogenesis of DKD including hyperglycaemia, altered haemodynamics and hyperlipidaemia as well as recently identified the gut-kidney axis. Moreover, the authors have provided a summary from both preclinical models of DKD and clinical trials using resistant starch as a dietary therapy in the management of DKD. The review article is well written with categorically distinct subheading to make it clear and easily understandable.
One minor issue is that the Food Sources for resistant starch type 4 and 5 are missing in Table 1.
Author Response
Reviewer 2:
The review article has discussed the mechanisms involved in this pathogenesis of DKD including hyperglycaemia, altered haemodynamics and hyperlipidaemia as well as recently identified the gut-kidney axis. Moreover, the authors have provided a summary from both preclinical models of DKD and clinical trials using resistant starch as a dietary therapy in the management of DKD. The review article is well written with categorically distinct subheading to make it clear and easily understandable.
One minor issue is that the Food Sources for resistant starch type 4 and 5 are missing in Table 1.
Thank you for the time taken to review our manuscript. RS4 is formed by chemical modification, whilst RS5 form is formed through a starch-lipid complex interaction although recently other complexes, such as starch-protein complexes, have been identified. There are no natural food sources of these RS types (e.g. RS4 could be formed by chemical modification and then added to a processed food product). We hope this explanation provides clarity on this point.
Reviewer 3 Report
Nutrients Peer Review
Authors have done a comprehensive review of the literature
The tables and figures complement the text
Suggestions to strengthen the manuscript
Table 3
Suggest expansion column 1 (ref #) – expand to author, date, country
Change from People to “humans”
Conclusions
Expand to translate limitations of present review into 3-4 clear research action statements such as need for food tables (lines 381) necessary to estimate habitual consumption to align/compare with RCT
Table 2
See above; description of RS levels does not help reader understand quantity to consume in normal foods; for example, R1 – quantity of legumes to equal a serving comparable to whole grains. Help connect concept with practitioner application. What should a practitioner tell a patient at the present time in layman terms?
Author Response
Reviewer 3:
Nutrients Peer Review
Authors have done a comprehensive review of the literature
The tables and figures complement the text
Suggestions to strengthen the manuscript
Table 3
Suggest expansion column 1 (ref #) – expand to author, date, country
Thank you for this suggestion. In the interest of formatting to keep Table 3 on one single page, we have not included this extra information.
Change from People to “humans”
We have changed the Table name to:” Table 3. Overview of Studies Investigating Resistant Starch In Humans With Chronic Kidney Disease And Diabetic Kidney Disease.“
Conclusions
Expand to translate limitations of present review into 3-4 clear research action statements such as need for food tables (lines 381) necessary to estimate habitual consumption to align/compare with RCT
We have included a section on Future Directions (Lines 532-541) to provide a clear overview of the need for food databases of RS.
Table 2
See above; description of RS levels does not help reader understand quantity to consume in normal foods; for example, R1 – quantity of legumes to equal a serving comparable to whole grains. Help connect concept with practitioner application. What should a practitioner tell a patient at the present time in layman terms?
We have included reference to foods with high concentrations of RS in lines 292-294.